# Comparisons of Emotional Responses, Flow Experiences, and Operational Performances in Traditional Parametric Computer-Aided Design Modeling and Virtual-Reality Free-Form Modeling

**Yu-Min Fang *** and **Tzu-Lin Kao**

Department of Industrial Design, National United University, Miaoli 36003, Taiwan; aa861219aa@gmail.com
*   Correspondence: fangeo@nuu.edu.tw; Tel.: +886-37-38-1664

**Abstract:** Three-dimensional (3D) computer-aided design (CAD) is a vital tool for visualizing design ideas. While conventional parametric CAD modeling is commonly used, emerging virtual reality (VR) applications in 3D CAD modeling require further exploration. This study contrasts the emotional response, flow experience, and operational performance of design novices using VR free-form modeling (Gravity Sketch 3D) and conventional parametric CAD modeling (SolidWorks). We arranged two representative tasks for 30 participants: modeling an exact geometric shape (a cube) and a creative shape (a mug). We measured emotional response and flow experience through scales, and gathered operational performance, and further insights through semistructured interviews. The findings reveal more positive and intense emotional responses to VR free-form modeling, but its overall flow experience did not exceed expectations. No significant differences were found in concentration, time distortion sense, or control between the two techniques. Comparing modeling tasks, VR free-form modeling showed promising operational performance for early ideation, whereas conventional parametric CAD modeling proved to be more effective in 3D digitization of known shapes.

**Keywords:** virtual reality; computer-aided design; parametric CAD modeling; emotional valence and arousal; flow experience; immersion; operational performance

## 1. Introduction

Various industries continuously seek to optimize their processes and to enhance user experiences. The persistent pursuit of efficiency, accuracy, and creativity has been a significant driver for technological innovations. Among these innovations, virtual reality (VR), a technology capable of creating a three-dimensional (3D) virtual world that provides users with multiple sensory simulations, has been recognized for its vast potential.

In a VR environment, users feel as if they are actually at a real scene and can observe the scene in real time [1,2]. With the increase in demand for VR, coupled with the advances in computer calculation and the popularity of technology, VR has been applied in various industries and scenarios [3]. VR is widely used in various fields, such as games, medical care, training, and design. Among these fields, VR technology is highly applicable to industrial design.

The 3D computer-aided design (CAD) method enables designers to present their ideas visually. Conventional 3D CAD modeling is performed using a fixed parametric model, that is, using predefined parameters and constraints to create a model. This technique increases the productivity, efficiency, and accuracy in the design process [4,5] but it is sometimes limited by complexity and hinders creativity.

Free-form 3D modeling is another 3D CAD technique that is under development. This technique enables designers to create models by using aspects such as gestures and voice

and to produce complex and organic shapes in an easy and flexible manner. Free-form 3D modeling can be incorporated with VR technology to achieve higher modeling creativity and operational freedom. Compared with conventional 3D CAD modeling, VR modeling enables designers to view and operate a 3D model in real time, enabling the visualization of more comprehensible spatial relationships and design proportions. In brief, VR adds a more immersive and deeply interactive experience to CAD modeling.

In CAD modeling, designers' emotional responses, concentration, and sense of control are key factors that affect the design quality and overall design team productivity. Studies have explored the effect of CAD modeling on its users. However, scant research has compared the effects of free-form modeling and parametric CAD modeling on users' emotional responses, flow experiences, and operational performances, particularly, with the incorporation of VR technology.

Although the incorporation of VR brings a sense of novelty and enjoyment, no clarity exists regarding whether it enhances the concentration and sense of control of designers or whether it is advantageous for achieving the precision required for products designed through CAD, free-form modeling, and parametric modeling.

In this study, we compared the effects of VR free-form modeling and parametric CAD modeling on designers' emotional responses, flow experiences, and operational performances to clarify the strengths and weaknesses of the two techniques in terms of designers' states of mind. The objectives of this study are described as follows:

1. To explore the different operational characteristics of CAD modeling and to compare the effects of parametric CAD modeling and VR free-form modeling on designers.
2. To measure emotional response and flow experience and compare the emotional valence, emotional arousal, and flow experience resulting from the use of parametric CAD modeling and VR free-form modeling.
3. To provide recommendations for the CAD process in industrial design and to determine the potential strengths and weaknesses of parametric CAD modeling and VR free-form modeling.

## 2. Literature Review

### 2.1. Development of VR Devices and Software

VR is a 3D virtual world generated through computer simulation. It provides users with multiple sensory simulations and a sense of presence, which enable them to observe the surroundings in 3D space in real time and without limitation [6]. Krueger regarded VR as an interface with three features, namely immersion, imagination, and interaction [7]. Immersion refers to a condition in which users are completely immersed in a scenario to complete a task or goal and they feel delighted and satisfied that they have forgotten about the real world. Imagination refers to a condition in which sensory stimuli create a virtual environment and convince users that they exist in the virtual space. Finally, interaction refers to a condition in which users can travel freely and interact with the surroundings in the virtual space [8].

In VR applications, users can interact with the virtual objects in the scene by using various sensing devices, including 3D glasses and gloves. When users move, the computer can immediately perform complex calculations and present an exact, timely 3D image, thereby creating a sense of presence [9]. VR displays can be employed to experience VR. Many companies have developed and launched VR displays with different features. Currently available professional head-mounted displays include Sony PlayStation VR, HTC Vive, and Oculus Rift, all of which are high-end products that must be attached to VR-compatible computers or equipment with a high computing speed.

The use of VR and extended reality (XR, representing all technologies that merge the physical and virtual worlds) has been widely investigated across various sectors, notably in Industry 5.0, Architecture, Engineering, and Construction (AEC), and education. Tu et al. (2023) emphasized the role of digital twins (DTs) and XR in accelerating the human-centric Industry 5.0 transformation, highlighting their ability to provide digital

representations of physical assets and offer interactive environments for humans and machines [10]. Alizadehsalehi et al. (2020) demonstrated the value of XR in simulating multidimensional digital models of construction projects [11], while Bellalouna (2019) discussed the potential of VR for cognitive design-review and failure modes and effects analysis (FMEA) in industrial engineering [12].

VR integration with Building Information Modeling (BIM) has been recognized for facilitating project integration [13], while VR-based engineering education was proposed as a method to enhance manufacturing sustainability in Industry 4.0 [14]. Havard et al. (2019) proposed that DTs and VR are key for the design and optimization of cyber-physical production systems within Industry 4.0 [15], and Alizadehsalehi and Yitmen (2023) explored the intersection of BIM, DTs, and XR for automated construction progress monitoring [16]. In summary, VR and XR technologies hold the potential to transform various areas of our lives, from the design and construction process to human–machine interaction and modern manufacturing practices.

### 2.2. CAD, Parametric Modeling Software, and VR-Technology-Based Learning

In CAD, the design idea is presented using paper and pen and in a 3D form, which makes the entire process of creativity, design, and manufacturing intuitive and simple. This characteristic of CAD also helps to improve the spatial ability of design beginners during their learning phase [17]. For example, the use of 3D CAD software can improve individuals' positioning abilities in 3D rotation and their spatial organizational abilities in transformation from a two-dimensional (2D) space to a 3D space [18].

The parametric modeling technique involves the use of parametric feature-based methods to create models and components. The adopted parameters can be numerical or geometric parameters, which are easy to edit and modify. The aforementioned technique increases design efficiency and accuracy [19]. SolidWorks is a 3D CAD modeling software program developed by the SolidWorks Corporation of Dassault Systèmes SE (Vélizy-Villacoublay, France). This software runs on Microsoft Windows and has widespread application. As of 2013, more than two million engineers and designers from over 165,000 companies worldwide have used SolidWorks [20]. In the present study, we employed SolidWorks as the experimental software for CAD parametric modeling.

In design learning methods based on conventional CAD and drafting models, 2D concepts are used to introduce 3D spatial concepts to learners. However, problems occur when such a learning method is applied. For example, design beginners must rely on their imagination to conceive objects in a 3D space. In addition, the creation of 3D graphics requires considerable time. The application of VR technology in 3D design learning is receiving increasing attention. A study by [21] revealed that VR technology could solve the aforementioned problems and provided a highly intuitive and interactive learning experience, which stimulated the imagination of design beginners. Therefore, the application of VR in 3D design teaching has considerable potential. VR can provide a rich interactive experience and intuitive sensory stimulation, thereby facilitating design beginners' understanding and experiences of the 3D design and modeling process. When learning to use CAD software, design beginners can easily comprehend the design and manufacturing process through the simulation and assembly experience offered in the VR environment. Moreover, VR can provide challenging scenarios and tasks, which stimulate the creativity and problem-solving skills of design beginners.

Accordingly, CAD, parametric modeling, and VR-based modeling are indispensable methods in the fields of modern design and manufacturing. In addition to increasing the efficiency and quality of product design, these techniques enhance the learning outcomes and spatial ability of design beginners and stimulate their creativity and problem-solving skills. Thus, outstanding talents can be cultivated for the design and manufacturing fields by using the aforementioned techniques.

### 2.3. Emotional Response and the Self-Assessment Manikin

Because of limitations related to experimental equipment and time, emotional response is typically measured using a self-report method. The most commonly used self-report scale is the self-assessment manikin (SAM), which consists of three dimensions, namely pleasure, arousal, and dominance [22,23]. The dominance dimension of this scale has been found to be unsuitable for measuring the emotional response to computer interfaces; thus, a revised 2D SAM was developed for measuring this response [24]. In the present study, this 2D SAM was employed for measuring emotional response. The items of the 2D SAM were measured using a 9-point Likert scale. The dimensions of the 2D SAM were related to emotional valence and emotional arousal, which are described as follows:

- Regarding emotional valence (valence and level of pleasure), researchers have interpreted emotional valence as being pleasurable or unpleasurable. Lang replaced the concept of pleasure with valence and divided valence into two extreme categories, namely positive and negative [25]. In the present study, we applied the concept of Lang to estimate participants' positive and negative emotional responses during CAD modeling.
- With respect to emotional arousal, in addition to positivity and negativity, emotion can be estimated in terms of intensity. Emotional arousal refers to the physical and mental changes caused by external stimuli. Individuals' facial expressions and limb postures tend to change with their thoughts, and their emotional intensity changes because of different stimuli. Emotional arousal can be measured using a scale from "calm" to "excited" [26,27]. The source of emotion is generally related to individuals' past experiences, whereas the source of emotional arousal is primarily associated with the current situations that they encounter. Therefore, the intensity of emotional arousal is usually the reaction caused by external situational stimuli.

### 2.4. Flow Experience

The concept of flow experience was proposed by Csikszentmihalyi and Csikszentmihalyi. When individuals are completely involved, they are focused on an activity and filter out all irrelevant perceptions, they enter a state of flow experience [28]. The most crucial feature of VR is the sense of presence that it provides, which makes the artificial environment appear real [29]. The sense of presence is a mediating mechanism for VR technology and an objective index for evaluating the authenticity of a VR system [30].

Regarding the measurement index for flow experience, studies have indicated that flow experience has a playful characteristic [31]. When VR users are completely engaged in an activity, they feel that the time is stagnant, consider matters irrelevant to the activity to be unimportant, and experience time distortion [32]. Moreover, researchers have employed control, concentration, and time distortion as indices for flow experience [33]. Accordingly, in the present study, we used four indices to measure flow experience: concentration, control, enjoyment, and sense of time distortion [34,35]. Concentration refers to focus, involvement, and the condition of not thinking about other matters and not being influenced by others. Control refers to the ability to manage and cope with any matter or change. Furthermore, enjoyment refers to fun, playfulness, excitement, and a sense of novelty. Finally, sense of time distortion refers to rapid time loss, the loss of the sense of time, and the forgetting of troubles and other matters. Questionnaire items related to these four indices were measured using a 5-point Likert scale.

### 3. Materials and Methods

#### 3.1. Research Design and Subjects

In this study, we compared the effects of conventional parametric CAD modeling (SolidWorks, SolidWorks Corporation of Dassault Systèmes SE, Vélizy-Villacoublay, France) and VR free-form modeling (Gravity Sketch 3D, Gravity Sketch Limited, London, the United Kingdom) on designers' emotional responses, flow experiences, and operational performances. Two representative tasks were designed for each operation technique,

namely modeling of an exact geometric shape (a cube) and modeling of a creative shape (a mug).

Before the experiment, we played an instruction and precaution video of approximately 10 min to the participants of this study and provided them with another 10 min for further exploration and queries. Then, they were asked to perform the two operational tasks. When using the VR free-form operative technique, the participants were asked to stand in the middle of the computer room in which the experiment was conducted. The examiners helped the participants to put on a VR head-mounted display, use a controller, and activate the Gravity Sketch 3D software on the Steam platform on a laptop. Subsequently, the participants began to work on the two tasks, and the examiners kept track of the elapsed time. After the participants completed the operational tasks, the examiners helped the participants take off the VR head-mounted display. Then, the participants were asked to fill out a questionnaire, and then they were invited to a semistructured interview. The experimental process and settings of VR free-form modeling are displayed in Figure 1. When using the conventional parametric CAD modeling technique, the participants were asked to sit in front of a desktop in the computer room, operate the SolidWorks 3D modeling software, and perform the two operational tasks under the examiners' instructions. The subsequent experimental procedures were the same as those in the VR free-form modeling. The experimental process and settings of conventional parametric modeling are depicted in Figure 2.

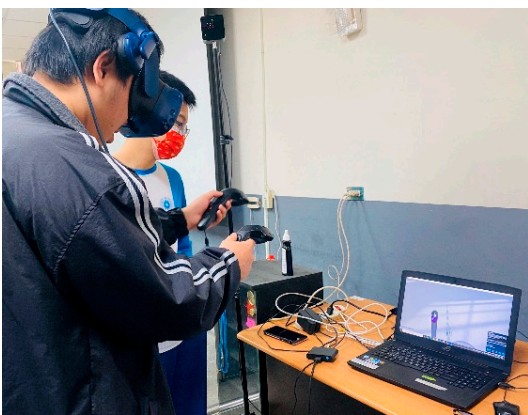

**Figure 1.** Experimental process and settings of virtual-reality (VR) free-form modeling.

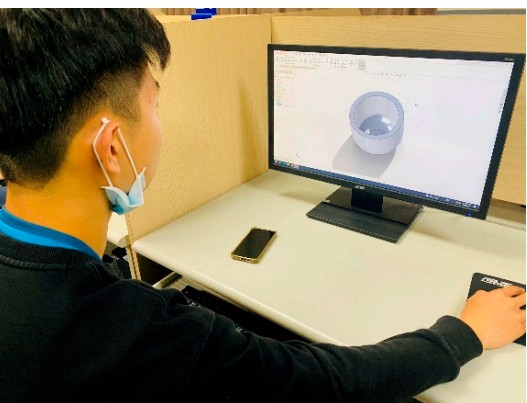

**Figure 2.** Experimental process and settings of conventional parametric computer-aided design (CAD) modeling.

We speculated that experienced designers have unique preferences for modeling software; therefore, we recruited design beginners as participants. A total of 30 third-grade vocational high school students participated in this study. Since the experiment was

incorporated as part of the course curriculum, all participants completed the test. Among these students, half of the students were male students, and half of the students were female students, with the mean age of the students being 17 years. A total of 24 students had experience in using VR products (80.0%); however, 73.3% of the students had a "moderate" or lower frequency of use, which might be related to the low prevalence of VR. Furthermore, six students had a very low frequency of computer use (20.0%), and only three students had never used 3D modeling software (10.0%).

### 3.2. Materials

In this study, two stimuli (modeling techniques) were designed: conventional parametric CAD modeling and VR free-form modeling (see Table 1). Parametric CAD modeling has been extensively used in different fields. Because of its widespread usage, the SolidWorks modeling software (2016 version) was employed for the parametric CAD modeling in this study. ASUS computers equipped with an NVIDIA discrete graphics processing unit and a 20-inch, light-emitting-diode screen were used in the parametric CAD modeling experiment. The computers were located in the computer room of a school.

**Table 1.** Descriptions of the conventional parametric CAD modeling and VR free-form modeling.

| Item | Parametric CAD | Free-Form VR |
|---|---|---|
| Operative technique | Desktop | VR devices |
| Hardware | ASUS desktop computer, LED display screen, keyboard, and mouse. | HTC VIVE Pro, a handheld controller, a base station, and an Acer VR-ready laptop |
| Specifications | Display: 20-inch, light-emitting-diode (LED) screen<br>Maximum resolution: 1366 × 768 pixels<br>Viewing angle: (horizontal and vertical) 90 × 65 degrees<br>Graphics processing unit: The desktop computer equipped with an NVIDIA discrete graphics processing unit | Display: Dual AMOLED 3.5″ screens<br>Resolution: 1440 × 1600 pixels per eye (total 2880 × 1600 pixels)<br>Field of View (FOV): Approximately 110 degrees<br>Refresh Rate: 90 Hz |
| Software | SolidWorks 2016<br>(executed on Microsoft Windows) | Gravity Sketch 3D VR<br>(activation of SteamVR on Microsoft Windows) |
| Operation site |  |  |
| Experimental site | Computer room at a vocational high school | |
| Description | A desktop computer is used to execute the parametric modeling software and complete the designated modeling tasks | VR devices are used to execute the VR free-form modeling software to complete the designated modeling tasks |
| Operational task | The following tasks are completed:<br>1. The modeling of an exact geometric shape, namely a cube<br>2. The modeling of a creative shape, namely a mug | |

**Table 1.** *Cont.*

| Item | Parametric CAD | | Free-Form VR | |
|---|---|---|---|---|
| Screenshot of the completed product | 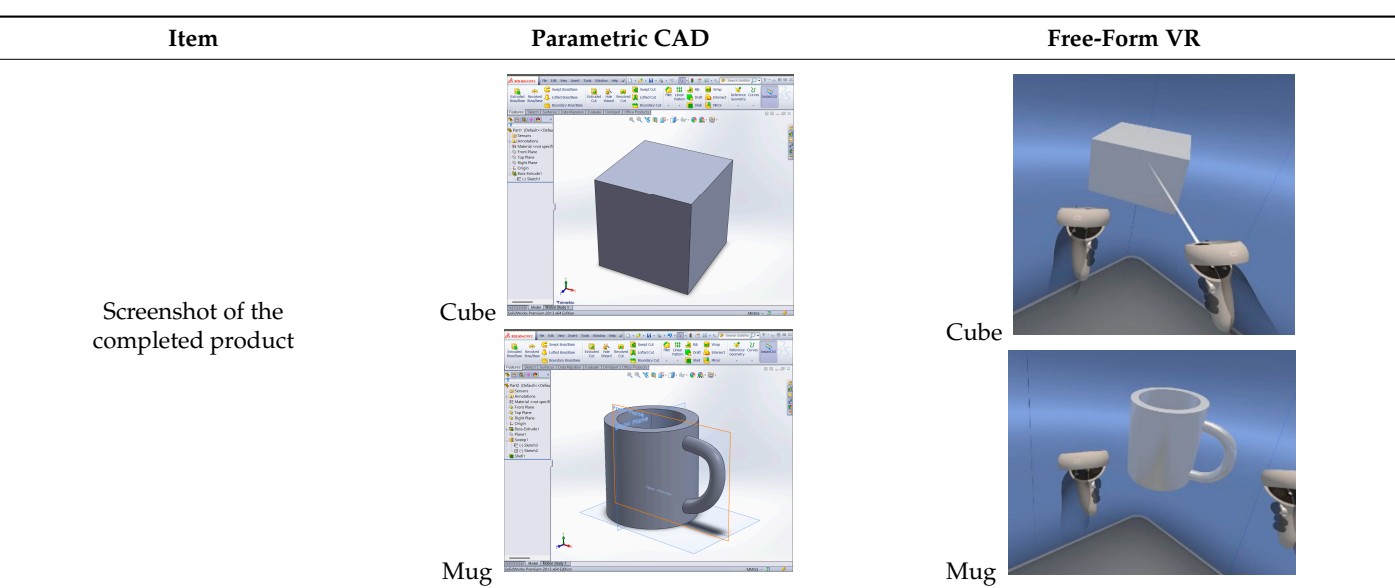 Cube | | Cube | |
| | Mug | | Mug | |

VR free-form modeling is an emergent modeling technique, and Gravity Sketch 3D was used as the software for VR free-form modeling in this study. In this creative 3D modeling software, a polygon modeling technique is used to connect points, edges, planes, and entire elements. Gravity Sketch 3D has features such as a 3D space brush, color selection system, and VR modeling controller, and this software enables designers to model their products by using a handheld VR controller. During testing, the participants put on a head-mounted display (HTC Vive, HTC and Valve Corporation) and used a wireless handheld controller (SteamVR, Valve Corporation) for operation. The examiners activated the motion program of SteamVR on a 15-inch, VR-ready Acer laptop with the Microsoft Windows operating system to monitor the experimental process.

The participants had to complete two modeling tasks in each modeling technique: (1) the modeling of an exact geometric shape, namely a cube, and (2) the modeling of a creative shape, namely a mug. A cube is a basic geometric shape with simple and symmetrical characteristics and its modeling requires precise size control. A mug has a free, creative shape, and its modeling process is more complex than that of a cube.

*3.3. Questionnaire Design*

To ensure that the questionnaire items conformed to the research purpose, we invited five experts to participate in the pretest and revised the questionnaire according to their opinions. The revised questionnaire is presented in Table 2. The questionnaire was divided into six sections, namely those related to emotional response, flow experience, operational performance, the strengths and weaknesses of the operative techniques, personal information, and the semistructured interview. With regard to the first section, the SAM was used to estimate the participants' emotional valence and emotional arousal by measuring their levels of pleasure and emotional intensity, respectively. Flow experience was assessed in terms of the sense of presence that the interfaces provided for the participants in terms of four dimensions: concentration, enjoyment, sense of time distortion, and control. Operational performance was assessed in terms of the time required to complete the two operative techniques. The participants' thoughts on the strengths and weaknesses of the operative techniques were recorded. Moreover, the participants' personal information was collected to understand their backgrounds.

**Table 2.** Questionnaire content.

| Content | Number of Items | Item Type | Purpose |
|---|---|---|---|
| 1. SAM | | | To examine emotional valence and emotional arousal |
| 1.1 Emotional valence<br>1.2 Emotional arousal | 2 | 9-point Likert scale | 1.1 To determine the level of pleasure<br>1.2 To determine the level of emotional intensity |
| 2. Flow experience<br><br>2.1 Concentration<br>2.2 Enjoyment<br>2.3 Sense of time distortion<br>2.4 Control | 14 | 5-point Likert scale | To understand the sense of presence provided by the operative techniques<br>2.1 To examine focus, involvement, and the condition of not being influenced by others<br>2.2 To examine fun, playfulness, excitement, and a sense of novelty<br>2.3 To examine rapid time loss, the loss of the sense of time, and the forgetting of troubles and other matters<br>2.4 To examine the ability to control and cope with changes |
| 3. Operational performance | 1 | Value (s) | To understand the time required for performing different operative techniques (recorded by the examiners) |
| 4. Strengths and weaknesses | 2 | Short-answer questions | To understand the participants' opinions on the strengths and weaknesses of the operative techniques |
| 5. Personal information | 7 | Multiple-choice questions | To understand the participants' personal background |
| 6. Interview | 1 | Semistructured interview | To understand the participants' thoughts on the operative techniques |

The questionnaire was developed and disseminated through Google's online survey platform. Figure 3 presents Question 1-1 as an illustrative example. Participants are required to adhere to the provided instructions and sequentially fill in the checkboxes. Progression to subsequent pages was only possible once the current page's responses had been duly completed.

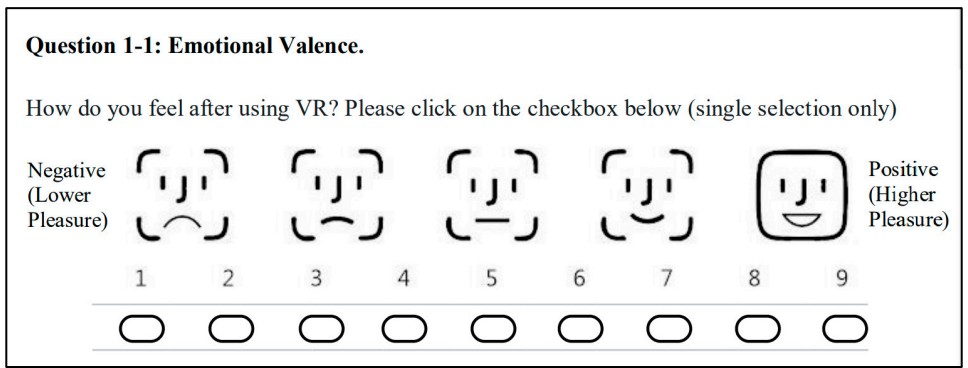

**Figure 3.** Google online questionnaire (Question 1-1 presented as an example).

In addition to filling out the formal questionnaire, the participants were asked to participate in a semistructured interview to clarify the results of the questionnaire survey. The interview was divided into three parts: (1) inquiring about the participants' preferences for the operative techniques, (2) inquiring about the participants' recommendations regarding the revision of the interfaces used in the two operational tasks (cube and mug modeling) and the reasons for these recommendations, and (3) inquiring about the participants' thoughts on their emotional responses, flow experiences, and operational performances.

Finally, the collected data were analyzed using SPSS 20.0. The primary statistical analysis involved descriptive statistical analysis and independent samples *t*-tests.

## 4. Results

### 4.1. Analysis of Emotional Responses

The participants' emotional responses to conventional parametric CAD modeling and VR free-form modeling were analyzed. After the participants completed the two designed tasks, they were asked to fill out the SAM, which consisted of the emotional valence and emotional arousal dimensions. Emotional valence was estimated in terms of pleasure levels by using a 9-point Likert scale, with a higher score indicating a higher pleasure level. Emotional arousal refers to a participant's emotional intensity and was also measured using a 9-point Likert scale, with a higher score indicating a higher emotional intensity. In the statistical results, the mean values of emotional valence and emotional arousal, as well as their corresponding line charts, standard deviations, and *t*-test results for both conventional parametric CAD modeling and VR free-form modeling are elaborated as shown in Figure 4 and Table 3.

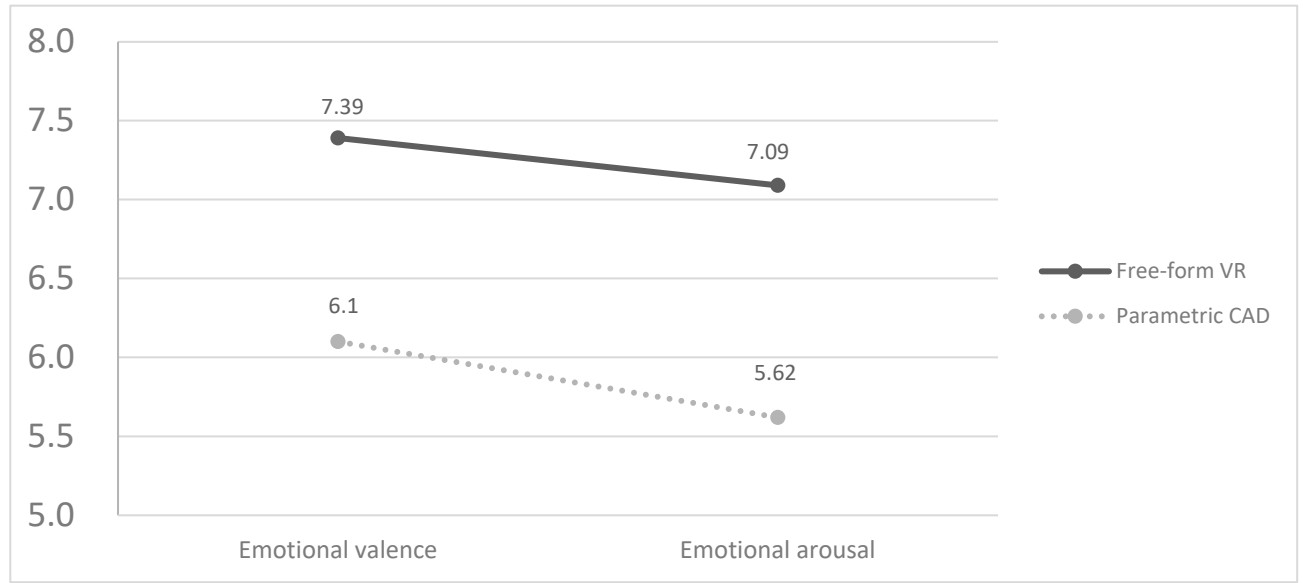

**Figure 4.** Emotional responses generated by conventional parametric CAD modeling and VR free-form modeling (unit, points; nine-point Likert scale).

**Table 3.** Means, standard deviations, and *t*-test results obtained for the emotional valence and emotional arousal produced by conventional parametric CAD modeling and VR free-form modeling (unit, points; nine-point Likert scale; * $p < 0.05$).

| Emotional Response | | Mean | Standard Deviation | *t*-Test on the Mean | |
| --- | --- | --- | --- | --- | --- |
| | | | | *t* | Significance (Two-Tailed) |
| Emotional valence | Free-form VR | 7.39 | 2.15 | 2.10 | 0.04 * |
| | Parametric CAD | 6.10 | 1.92 | | |
| Emotional arousal | Free-form VR | 7.09 | 2.02 | 2.74 | 0.01 * |
| | Parametric CAD | 5.62 | 1.47 | | |

Regarding emotional valence, the results indicated that VR free-form modeling (Gravity Sketch 3D) provided the participants with a higher level of pleasure (mean = 7.39, standard deviation = 2.15) than that provided by conventional parametric CAD modeling (SolidWorks) (mean = 6.10, standard deviation = 1.92). Although a significant difference was observed between the level of pleasure resulting from the two operative techniques ($p$ = 0.04), the means of both techniques exceeded 7, which suggested that both techniques stimulated positive emotions. However, we note that the significant difference is merely

0.04, thus, necessitating caution when interpreting these results. The close proximity of the *p*-value to the threshold implies a relatively weak rejection of the null hypothesis. Therefore, the interpretation of results should consider the substantive context and be corroborated with other evidence.

Regarding emotional arousal, the results suggested that VR free-form modeling generated a higher level of emotional arousal (mean = 7.09, standard deviation = 2.02) than that generated by conventional parametric CAD modeling (mean = 5.62, standard deviation = 1.47). A significant difference was observed between the level of emotional arousal resulting from the two techniques (*p* = 0.01), which implied that the participants experienced more intense emotions when using the VR free-form modeling technique than when using the parametric CAD modeling technique.

*4.2. Flow Experience: Level of Fun and Playfulness*

We employed four indices to measure flow experience: concentration, control, enjoyment, and sense of time distortion. These indices were measured on a 5-point Likert scale, with a higher score indicating a higher level of positivity. With regard to level of fun and playfulness, significant differences were observed between parametric CAD modeling and VR free-form modeling (*p* = 0.04 and *p* = 0.01, respectively). The mean values obtained for the four indices when using VR free-form modeling were higher than those when using parametric CAD modeling. The mean values of level of fun and playfulness were 4.00 (standard deviation = 1.12) and 3.95 (standard deviation = 1.06), respectively, when using VR free-form modeling and 3.38 (standard deviation = 0.80) and 3.19 (standard deviation = 0.98), respectively, when using parametric CAD modeling. In the statistical analysis, we present the mean values, line charts, standard deviations, and *t*-test results associated with the levels of fun and playfulness for both conventional parametric CAD modeling and VR free-form modeling, as illustrated in Figure 5 and detailed in Table 4.

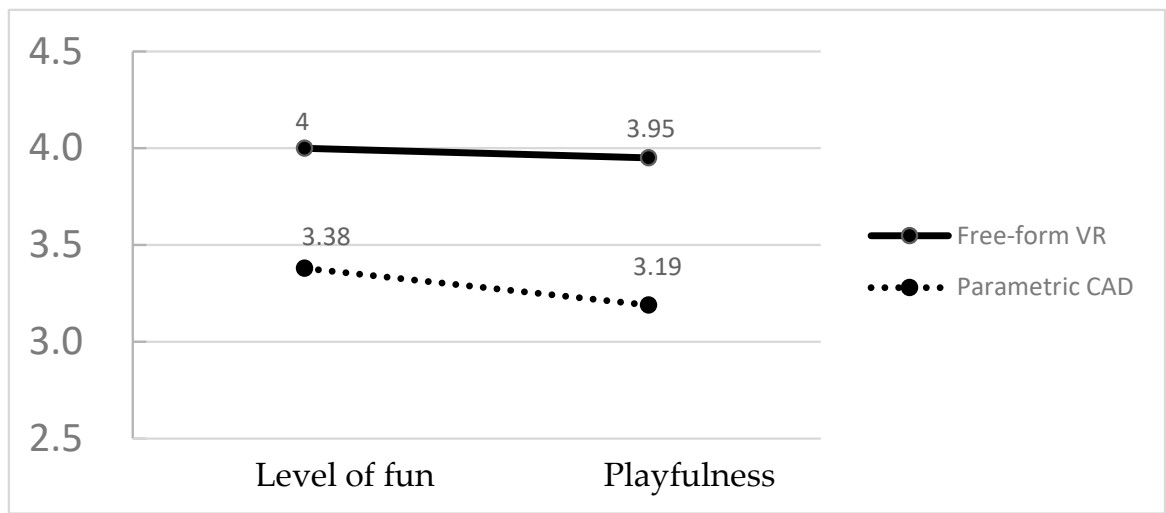

**Figure 5.** Mean levels of fun and playfulness when performing conventional parametric CAD modeling and VR free-form modeling (unit, points; five-point Likert scale).

However, the data reveal only a marginally significant difference in the perceived level of fun, noted at 0.04. Given the close proximity of the *p*-value to the significance threshold, this necessitates a cautious interpretation. It is crucial to contextualize these findings and validate them against other evidence. Notably, when taking into account overall immersion and its indicators—focus, time distortion, and control—no significant differences emerge. These aspects, in conjunction with the borderline *p*-value for perceived interest, call for a comprehensive examination.

**Table 4.** Mean values, standard deviations, and *t*-test results for level of fun and playfulness when performing conventional parametric CAD modeling and VR free-form modeling (unit, points; five-point Likert scale; * $p < 0.05$).

| Enjoyment | | Mean | Standard Deviation | *t*-Test on the Mean | |
|---|---|---|---|---|---|
| | | | | *t* | Significance (Two-Tailed) |
| Level of fun | Free-form VR | 4.00 | 1.12 | 2.07 | 0.04 * |
| | Parametric CAD | 3.38 | 0.80 | | |
| Playfulness | Free-form VR | 3.95 | 1.06 | 2.47 | 0.01 * |
| | Parametric CAD | 3.19 | 0.98 | | |

### 4.3. Operational Performance

During the experiment, the time required to complete the two modeling tasks was recorded, and the operational performances of VR free-form modeling and parametric CAD modeling in the tasks was compared. In the statistical analysis, we present the mean values, line charts, standard deviations, and *t*-test results for the operational performance for the two tasks (cube and mug) when performing conventional parametric CAD modeling and VR free-form modeling, as illustrated in Figure 6 and detailed in Table 5.

According to the results, higher operational performance was achieved in VR free-form modeling than in conventional parametric CAD modeling. The mean time required for the cube modeling task performed using VR free-form modeling was only 8.73 s (standard deviation = 9.21), which was shorter than that required for the cube modeling task performed using parametric CAD modeling (104.40 s, standard deviation = 181.52). The difference in operation performance between the two methods for the cube modeling task was significant ($p = 0.006$). The mean time required for the mug modeling task performed using VR free-form modeling was 33.43 s (standard deviation = 59.60), which was shorter than that required for the mug modeling task performed using parametric CAD modeling (479.43 s, standard deviation = 257.03). The difference in operation performance between the two methods for the mug modeling task was significant ($p = 0.000$). However, both datasets exhibit large standard deviations, indicating the presence of other complex variables that could be causing interference.

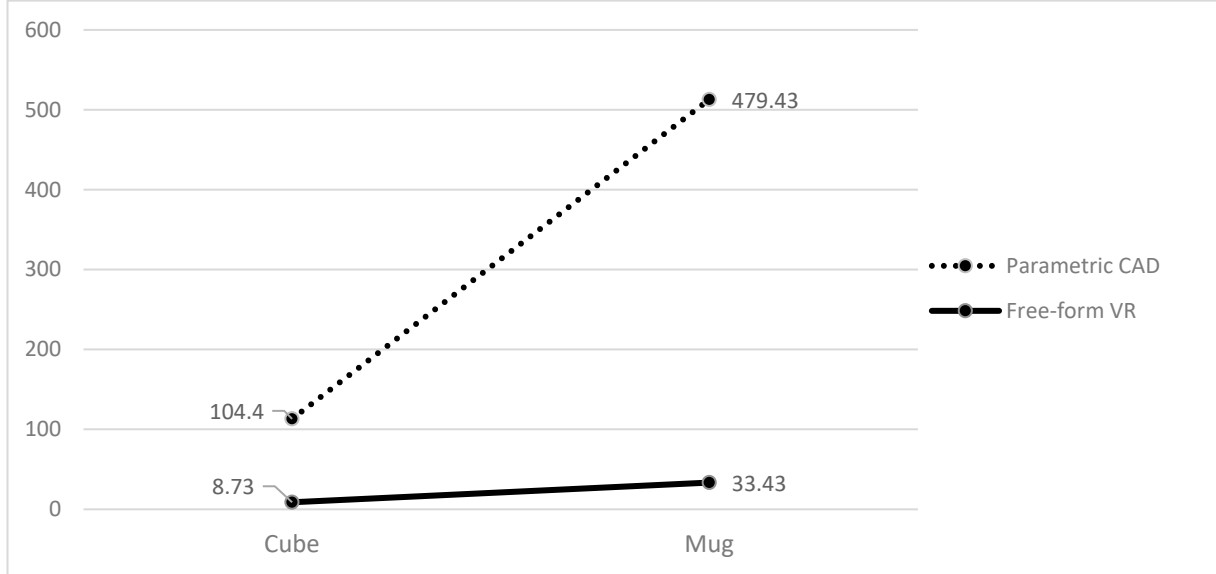

**Figure 6.** Operational performance for the two tasks when performing conventional parametric CAD modeling and VR free-form modeling (unit, second).

**Table 5.** Mean values, standard deviations, and *t*-test results for the operational performance for the two tasks when performing conventional parametric CAD modeling and VR free-form modeling (unit, second; * $p < 0.05$).

| Operational Performance | | Mean | Standard Deviation | t-Test on the Mean | |
|---|---|---|---|---|---|
| | | | | *t* | Significance (Two-Tailed) |
| Cube | Free-form VR | 8.73 | 9.21 | −2.88 | 0.006 * |
| | Parametric CAD | 104.40 | 181.52 | | |
| Mug | Free-form VR | 33.43 | 59.60 | −9.25 | 0.00 * |
| | Parametric CAD | 479.43 | 257.03 | | |

## 5. Discussion

### 5.1. Main Findings

In terms of emotional response, VR free-form modeling led to higher levels of emotional valence and emotional arousal than those of conventional parametric CAD modeling. According to the interviews, the incorporation of VR provided the participants with a sense of novelty and an increased level of operational freedom, which stimulated more positive and intense emotions in them. However, as previously mentioned, we noticed that the significant difference in emotional valence was only 0.04, with the *p*-value nearing the threshold. This calls into question whether other research methods might offer more compelling evidence. For instance, other scholars have reported state-of-the-art related studies. Some have confirmed that virtual embodiment intensified the emotional response to virtual stimuli [36]. If we view the operation of CAD as an extension of the individual, we find results consistent with this study, affirming that VR can indeed enhance emotional responses. Furthermore, the literature suggests alternative measurement methods. For instance, electrodermal activity (EDA) has been used to measure physiological responses of users to VR [37], which could be cross-verified with the SAM scale used in this study. These diverse studies can serve as references for future research expanding on emotional responses.

On the basis of the results of flow experience, we speculated that the participants had a sense of presence when using the VR product and felt fun and experienced playfulness during their timely interaction with the surroundings in the digital world. In addition, the participants mentioned that, when executing Gravity Sketch 3D, they felt that they had entered another time and space. The VR-based learning model implemented in this study was different from the learning model that the participants were used to; therefore, they experienced increased exuberance when performing VR free-form modeling.

Regarding other indices of flow experience, no significant differences were observed in concentration, sense of time distortion, and control between the two techniques. This result suggested that VR did not lead to a higher sense of time distortion, concentration level, and sense of control during operations. The existing literature on flow experience largely pertains to gaming or educational research [38,39], and a common conclusion drawn from these studies has been that the use of VR could evoke a higher degree of immersion. However, the findings of this study diverge from such conclusions. It appears that once the initial novelty and enjoyment brought by VR are set aside, VR does not live up to its expectations when transitioning into practical work scenarios.

In the interviews, the participants who used parametric CAD modeling implied that they lost their sense of direction and space when modeling a precise 3D graphic on a 2D screen; thus, they felt frustrated. Moreover, because the participants had little intuitive interaction with the models, they experienced a relatively low level of fun; therefore, they exhibited low scores in the fun dimension.

In accordance with the results of the operational performance, we speculated that the participants obtained more benefits when using VR free-form modeling than when using parametric CAD modeling because VR free-form modeling provided higher intuitiveness.

In an unfamiliar environment, the participants could still determine the correct approach for conducting VR free-form modeling; therefore, they exhibited higher operational performance in VR free-form modeling than in parametric CAD modeling. In particular, in the mug modeling task, the ability to construct free-form surfaces made the modeling process more comprehensive. However, the interviews revealed that the accuracy of cube modeling was relatively low because VR free-form modeling lacked a strict datum setting and precise numerical inputs.

The participants also mentioned that they struggled with the limited interface prompts and operation steps in parametric CAD modeling. For those who used parametric CAD modeling for the first time, the operation was initially difficult. Some encountered errors and were unable to troubleshoot the problem, which resulted in considerable time wastage. However, with suitable training, users could quickly and precisely edit their product during parametric CAD modeling, which represents a strength of parametric modeling.

Furthermore, as previously mentioned, the operational performance displayed a high standard deviation. This can be attributed to factors identified through field observations and interviews, such as spatial awareness, knowledge of 3D modeling, and proficiency with the operations, which significantly influence the time taken to complete tasks. It is suggested that leveraging the distinct characteristics of free-form VR and parametric CAD could uncover their unique advantages, thereby facilitating more consistent operational performance. Detailed recommendations are outlined in the following section.

### 5.2. Practical Implications

Drawing upon the comprehensive analysis of the statistical results and the insights derived from the interviews, we propose the ensuing practical strategies and recommendations for the enhancement of parametric CAD modeling and VR free-form modeling in practical applications:

1. After decades of interface optimization, the interfaces of current parametric modeling software programs have become graphical ones. However, the rigorous parametric logic of these programs makes learning for design beginners difficult. The enjoyment and emotional response produced by VR technology can reduce design beginners' learning frustrations. Features such as intuitive operation and natural dialogue are recommended to be introduced into future CAD modeling software programs to increase learners' interests and reduce the learning threshold.

2. An advantage of VR interfaces is that they improve self-learning ability. However, in reality, design beginners might be unable to receive instant assistance in device adjustment or software operation. Therefore, developers of CAD modeling software are advised to provide self-learning courses to enable design beginners to familiarize themselves with VR operation without time pressure. Additional prompts and instructions should be provided in the learning process to assist design beginners.

3. Developers of CAD modeling software are recommended to increase the usability of VR devices to increase the ease of learning. For example, the complicated positioning calibration and setting procedures of these devices can be simplified. Furthermore, the popularity of VR devices has increased with decreases in their component size and cost. Software developers should create solutions to increase the usability of and ease of learning with VR devices.

4. The participants suggested that software developers could consider designing customized VR controllers with similar shape, weight, and operation to real pens. Such controllers could also be used to optimize parametric modeling.

5. Some design beginners reported that they experienced slight dizziness after the experiment. We recommend that users should avoid making continual body movements during VR CAD modeling. To reduce dizziness, functions such as viewing angle stabilization and an interface with fixed time and fixed space can be developed.

6. According to the examiners' observations, the participants who had experience in using VR devises exhibited smooth reactions, whereas those who had never used VR

devices exhibited rather unnatural reactions because they required time to familiarize themselves with the operation. The interface operation of those who had experience in 3D modeling was smoother than that of those who had no experience in 3D modeling. The semistructured interviews indicated that those with experience in 3D modeling easily understood 3D spatial concepts, whereas those without this experience required time to understand these concepts, which affected their operational performance. Although VR modeling has intuitive operation, design beginners should receive basic training on 3D spatial concepts, regardless of the modeling techniques that they adopt.

### 5.3. Conclusions

This study compared the emotional responses, enjoyment, and operational performances of design beginners using two techniques, namely parametric CAD modeling and VR free-form modeling. We discovered that the emotional response, enjoyment, and operational performance achieved with these methods were significantly different. The results of this study confirmed that the incorporation of VR into conventional CAD modeling was beneficial because it resulted in more positive and intense emotional experiences as well as higher enjoyment, which led to better operational performance. We formulated recommendations for improving VR free-form modeling on the basis of the obtained statistical and interview results. The conclusions of this study are as follows:

1. The participants who used VR free-form modeling exhibited higher levels of emotional valence and emotional arousal than those who used parametric CAD modeling. The incorporation of VR into CAD modeling brought a sense of novelty and operational freedom for design beginners, which enabled them to have positive and intense emotional responses.
2. VR free-form modeling was more interesting than conventional parametric CAD modeling; however, the overall flow experience produced by VR free-form modeling was not as high as expected. No significant differences were found in concentration, sense of time distortion, and sense of control between the two modeling techniques. Although VR can bring enjoyment and promote learning, in practice, designers are still faced with challenging problems, such as completing professional modeling work, to attain their goals. Under such circumstances, professional training, such as logical training in map construction and the acquisition of relevant skills, becomes the main factor influencing all operative techniques.
3. We compared the performance of the participants in two modeling tasks, namely basic shape modeling (cube modeling) and free-form surface modeling (mug modeling), and found that VR free-form modeling led to higher operational performance than conventional parametric CAD modeling. The intuitive operation of VR free-form modeling enabled the participants to learn from their mistakes, and this operation was more suitable for shaping free-form surfaces than that of parametric CAD modeling. However, because VR free-form modeling lacked strict logic and precise numerical inputs, the accuracy of cube modeling was relatively low. By comparison, parametric CAD modeling enabled rapid and precise model creation, and the modeling result could easily be edited. Accordingly, in practice, VR free-form modeling is suitable for early ideation, whereas conventional parametric CAD modeling can be used for the 3D digitization of known shapes.

### 5.4. Future Works

Future works of this study can be further expanded in the following aspects:

Subclassification of CAD drawing tasks: Currently, we have simplified the classification of CAD drawing tasks into basic modeling and complex surfacing. However, CAD drawing tasks are diverse and complex. Future research can analyze and classify different types of CAD drawing tasks in more detail, and investigate the performance and efficiency differences of VR free-form modeling and parametric CAD modeling in these specific tasks.

Focus on technological trends: Free-form VR technology and traditional parametric CAD interfaces are constantly evolving and improving. Future researchers need to closely monitor the latest trends in these technologies to understand their development directions and their impact on CAD modeling. Researchers can explore breakthroughs in human–machine interface design and interaction methods to enhance user experience and operational efficiency.

User feedback and design optimization: This study mentioned users' feedback and suggestions on the operation experience of VR and parametric CAD. Future researchers can further investigate user feedback and needs, and optimize the design of CAD modeling software based on this feedback. This includes improving the intuitiveness of the user interface, providing more learning resources and training content, and addressing issues and difficulties that users may encounter.

Multimodal interaction and integration: Future research can focus on multimodal interaction and integration, combining VR free-form modeling and parametric CAD modeling with other technologies and devices. This may include gesture recognition, voice control, virtual reality, and augmented reality, to provide a richer and more flexible CAD modeling experience.

In summary, future researchers can further expand the results and contributions of this study through more detailed analyses of CAD drawing tasks, with a focus on technological trends, user feedback and design optimization, as well as multimodal interaction and integration. These efforts will contribute to a better understanding and application of the value and potential of VR free-form modeling and parametric CAD modeling in practical applications.

**Author Contributions:** Conceptualization, Y.-M.F. and T.-L.K.; Methodology, Y.-M.F. and T.-L.K.; Formal analysis, T.-L.K.; Investigation, T.-L.K.; Resources, Y.-M.F.; Data curation, T.-L.K.; Writing—original draft, T.-L.K.; Writing—review & editing, Y.-M.F.; Supervision, Y.-M.F.; Funding acquisition, Y.-M.F. All authors have read and agreed to the published version of the manuscript.

**Funding:** This research was funded by the National Science and Technology Council of Taiwan, grant number MOST 111-2410-H-239-016-.

**Institutional Review Board Statement:** This study was approved by the Human Research Ethics Committee at National Cheng Kung University (Approval No.: NCKU HREC-E-109-223-2).

**Informed Consent Statement:** Informed consent was obtained from all subjects involved in the study.

**Data Availability Statement:** Deidentified participant data supporting the findings of this study are available for an indefinite period of time from the corresponding author. The data can be accessed by professionals for research purposes by contacting the corresponding author directly.

**Conflicts of Interest:** The authors declared no potential conflict of interest with respect to the research, authorship, and/or publication of this article.

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
