# Peer review of "Comparisons of Emotional Responses, Flow Experiences, and Operational Performances in Traditional Parametric Computer-Aided Design Modeling and Virtual-Reality Free-Form Modeling"

_applsci, doi:10.3390/app13116568_

Round 1

Reviewer 1 Report

In most fields of activity, there is a tendency to consider the novel achievements in terms of tools and technologies better in all regards than the traditional tools and technologies. This article approaches such an issue in the field of design and this direction has certain merits.

The article is clear and easy to read by a professional interested in the respective subject. The approach is relevant and comprehensive, investigating the differences between parametric Computer Aided Design modelling and Virtual-Reality free-form modelling from different perspectives. The manuscript is well-structured with respect to the standard format of a scientific article.

Most of the cited references are recent and all are relevant to the subject approached. One author cites three of his own works, but they are relevant to the subject.

The article is scientifically sound, the research objectives are well-formulated, the methods well-chosen and the results are reproducible (considering all the details given).

The conclusions are consistent with the presented results.

I have some suggestions for improvement:

- In the “Questionnaire Design” subchapter, it would be better to give some exact examples of the questions used.

- The t-Test was well-chosen, considering the small size of participants, but in the tables should be indicated the value of t-critic.

- It should be commented that most of the p-values are smaller than alpha=0.05, but not much smaller. For example, p-value = 0.04 in Table 3.

- It should be commented about the large value of standard deviation when discussing the operational performance.

- It should be indicated whether all participants managed to successfully accomplish the tasks.

Author Response

Dear Reviewer,

I would like to express my sincere gratitude for your insightful comments and for affording us the opportunity to enhance the quality of our paper. Attached herein are our detailed responses, addressing each point of your comments and explaining the corresponding modifications made to the manuscript.

Should you have any additional remarks or suggestions, I would be more than willing to receive them. I am committed to refining this paper to the best of my abilities and I deeply appreciate your invaluable assistance in this endeavor.

Best Regards,

Yu-Min Fang

Ph.D., Professor

Department of Industrial Design, National United University

Address: No. 1, Lien-Da, Kung-Ching Li, Miao-Li, Taiwan, 36003

Tel.: +886 9 20032778

E-mail address: FanGeo@nuu.edu.tw; geoffreyfang@hotmail.com

Reviewer 2 Report

The manuscript reports findings of an empirical investigation where emotional responses, flow experience and operational performance of study participants (design beginners) are compared. The study is based on VR-based Free-Form Modling in a CAD software. Generally, the study has potential. However, there are still a couple of aspects to consider in a revision of the manuscript. I would like to mention the following points:

1)      The argumentation in the introduction follows the logic that a technological innovation (VR) leads to application in various industry. Actually, the logic is the other way around: as the industries have potentials, new innovations are required. (see line 46/47: “Because of advances in computer calculation, VR has been applied in various industries and scenarios [3]”). I would change the introduction by beginning with the potentials of the industry (not of calculation power) that cause the application of VR.

2)      The state-of-the-art section on arousal could be expanded. There are quite many new existing studies showing the multidisciplinary potentials of measuring physiological responses leading to interpretation of emotions (e.g. https://doi.org/10.3389/fpsyg.2021.674179 and https://doi.org/10.1007/s42489-023-00137-7)

3)      Section 4 should definitely be re-structured. It is quite unusual for an empirical investigation to present the results in a discussion section. Moreover, you do not discuss your results with reference to the state-of-the-art literature. You should provide a results and a discussion section. In your discussion, you should show in how far your studies extends, backs up or even contradicts existing findings.

The discussion section requires a thorough revision, as it does not refer to state-of-the-art literature.

Author Response

(The authors gave the same response as above.)

Reviewer 3 Report

1. The abstract needs to be shorter, and more attention should be paid to the main keywords of the article in this section.

2. The introduction and literature review sections are good; they just need to be improved by adding information regarding building information modeling (BIM), extended reality technologies (XR), and VR. The authors need to review the most recent VR articles (2019, 2020, 2021, 2022, and 2023). You have to review all the main keywords of the article in these sections. In order to help with this task, I provided some literature that you can use in your introduction or literature review sections:

-TwinXR: Method for using digital twin descriptions in industrial eXtended reality applications

- From BIM to extended reality in AEC industry

-Virtual-reality-based approach for cognitive design-review and fmea in the industrial and manufacturing engineering

- Virtual reality-based cloud BIM platform for integrated AEC projects

-Virtual reality-based engineering education to enhance manufacturing sustainability in industry 4.0

-Digital twin and virtual reality: a co-simulation environment for design and assessment of industrial workstations

- Digital twin-based progress monitoring management model through reality capture to extended reality technologies (DRX)

- Rules and validation processes for interoperable BIM data exchange

- Interactions of Sustainability and BIM in Support of Existing Buildings

3. In the Table 1, it can be helpful to add the FOV and some other specifications of the VRs that are used.

4. All tables and figures need to describe in the text adequately.

5. The authors strongly suggested adding a "practical implications" section before the Conclusion section.

6. The authors strongly suggested adding a "Future Works" section at the end.

1. The abstract needs to be shorter, and more attention should be paid to the main keywords of the article in this section.

2. The introduction and literature review sections are good; they just need to be improved by adding information regarding building information modeling (BIM), extended reality technologies (XR), and VR. The authors need to review the most recent VR articles (2019, 2020, 2021, 2022, and 2023). You have to review all the main keywords of the article in these sections. In order to help with this task, I provided some literature that you can use in your introduction or literature review sections:

-TwinXR: Method for using digital twin descriptions in industrial eXtended reality applications

- From BIM to extended reality in AEC industry

-Virtual-reality-based approach for cognitive design-review and fmea in the industrial and manufacturing engineering

- Virtual reality-based cloud BIM platform for integrated AEC projects

-Virtual reality-based engineering education to enhance manufacturing sustainability in industry 4.0

-Digital twin and virtual reality: a co-simulation environment for design and assessment of industrial workstations

- Digital twin-based progress monitoring management model through reality capture to extended reality technologies (DRX)

- Rules and validation processes for interoperable BIM data exchange

- Interactions of Sustainability and BIM in Support of Existing Buildings

3. In the Table 1, it can be helpful to add the FOV and some other specifications of the VRs that are used.

4. All tables and figures need to describe in the text adequately.

5. The authors strongly suggested adding a "practical implications" section before the Conclusion section.

6. The authors strongly suggested adding a "Future Works" section at the end.

Author Response

(The authors gave the same response as above.)

Round 2

Reviewer 2 Report

The authors provided a revised version of the manuscript and a response letter addressing all points mentioned in the first round of the review. The changes lead to an improvement, and the argumentation in the response letter is sound. There is still some improvement potential in the discussion section to point out the new findings more clearly. However, the main messages can be seen. Against this background, I would like to recommend this manuscript for publication.

The authors provided a revised version of the manuscript and a response letter addressing all points mentioned in the first round of the review. The changes lead to an improvement, and the argumentation in the response letter is sound. There is still some improvement potential in the discussion section to point out the new findings more clearly. However, the main messages can be seen. Against this background, I would like to recommend this manuscript for publication.

Reviewer 3 Report

ACCEPT

ACCEPT